 AMERICAN SOCIETY FOR MICROBIOLOGY | Microbiology SPECTRUM

# The Cell–Cell Communication Signal Indole Controls the Physiology and Interspecies Communication of *Acinetobacter baumannii*

Binbin Cui,[a] Xiayu Chen,[a] Quan Guo,[a] Shihao Song,[a] Mingfang Wang,[a] Jingyun Liu,[b] ![ORCID] Yinyue Deng[a]

[a]School of Pharmaceutical Sciences (Shenzhen), Shenzhen Campus of Sun Yat-sen University, Sun Yat-sen University, Shenzhen, China
[b]Department of Stomatology, Zhengzhou Shuqing Medical College, Zhenzhou, China

Binbin Cui and Xiayu Chen contributed equally to this article. Author order was determined by their equal but gradated contributions for this paper.

**ABSTRACT** Many bacteria utilize quorum sensing (QS) to control group behavior in a cell density-dependent manner. Previous studies have demonstrated that *Acinetobacter baumannii* employs an N-acyl-L-homoserine lactone (AHL)-based QS system to control biological functions and virulence. Here, we report that indole controls biological functions, virulence and AHL signal production in *A. baumannii*. The biosynthesis of indole is performed by A1S_3160 (AbiS, *Acinetobacter baumannii* indole synthase), which is a novel indole synthase annotated as an *alpha/beta* hydrolase in *A. baumannii*. Heterologous expression of AbiS in an *Escherichia coli* indole-deficient mutant also rescued the production of indole by using a distinct biosynthetic pathway from the tryptophanase TnaA, which produces indole directly from tryptophan in *E. coli*. Moreover, we revealed that indole from *A. baumannii* reduced the competitive fitness of *Pseudomonas aeruginosa* by inhibiting its QS systems and type III secretion system (T3SS). As *A. baumannii* and *P. aeruginosa* usually coexist in human lungs, our results suggest the crucial roles of indole in both the bacterial physiology and interspecies communication.

**IMPORTANCE** *Acinetobacter baumannii* is an important human opportunistic pathogen that usually causes high morbidity and mortality. It employs the N-acyl-L-homoserine lactone (AHL)-type quorum sensing (QS) system, AbaI/AbaR, to regulate biological functions and virulence. In this study, we found that *A. baumannii* utilizes another QS signal, indole, to modulate biological functions and virulence. It was further revealed that indole positively controls the production of AHL signals by regulating *abaI* expression at the transcriptional levels. Furthermore, indole represses the QS systems and type III secretion system (T3SS) of *P. aeruginosa* and enhances the competitive ability of *A. baumannii*. Together, our work describes a QS signaling network where a pathogen uses to control the bacterial physiology and pathogenesis, and the competitive ability in microbial community.

**KEYWORDS** *Acinetobacter baumannii*, quorum sensing, indole, virulence, competitive fitness

It is well known that quorum sensing (QS) is used by many bacteria to coordinate communal behaviors in a cell density-dependent manner (1–3). To date, many kinds of QS signaling molecules have been identified. Among them, N-acyl-L-homoserine lactones (AHLs) are well characterized and employed by many Gram-negative bacteria (4–7). In addition, other kinds of QS signals, such as diffusible signal factor (DSF) family signals, autoinducer-2 (AI-2), 2-heptyl-3-hydroxy-4(1H)-quinolone (PQS), 2-heptyl-4-quinolone (HHQ), methyl 3-hydroxypalmitate, autoinducer-3 (AI-3), bradyoxetin and diketopiperazines, were also identified as being utilized by bacterial cells to communicate with each other (8–14). In recent years, increasing evidence has confirmed the important role of indole in both intraspecies signaling and interspecies

**Ad Hoc Peer Reviewer** ![ORCID] Jintae Lee, Yeungnam University

Address correspondence to Yinyue Deng, dengyle@mail.sysu.edu.cn.

The authors declare no conflict of interest.

communication. Indole regulates spore formation, cell division, plasmid stability, antibiotic tolerance, biofilm formation, motility, and virulence in both indole-producing and non-indole-producing bacteria (15–17). The biosynthesis process for indole was also identified in *Escherichia coli*, in which biosynthesis of indole is performed by TnaA (18, 19). Indole induces the expression of multidrug exporter genes through the two-component systems BaeSR and CpxAR in *E. coli* (20). It also affects antibiotic tolerance of *Pseudomonas fluorescens* through EmhR (21).

*Acinetobacter baumannii* is an opportunistic Gram-negative pathogen that usually causes pneumonia and infections in the bloodstream and urinary tract, resulting in high morbidity and mortality (22–24). Abuse of antibiotics has led to the high clinical drug resistance of *A. baumannii*, and carbapenem-resistant *A. baumannii* has been listed in the WHO's first-ever list of priority antimicrobial-resistant pathogens (25–26). Previous studies have shown that *A. baumannii* employs the AHL-type QS system, which consists of the AHL synthase AbaI and the transcriptional regulator AbaR, to regulate biological functions and virulence (27–30). AbaI produces AHL signals, and AbaR functions as a receptor protein for AHL signals. After AbaR binds to AHL signals, the complex binds to the promoter region of target genes and controls gene expression.

QS signals play important roles in microbial ecology through interspecies communications, in addition to their critical role in intraspecies signaling (8, 9, 31). A previous study showed that indole produced by the gut microbiota suppressed the expression of the virulence genes of the enteric pathogens enterohemorrhagic *E. coli* (EHEC) (17). Furthermore, indole is not only an extracellular biofilm signal for *E. coli* but also represses QS-regulated virulence factors, including pyocyanin, PQS, pyochelin and pyoverdine of *P. aeruginosa* (32–34). As an interspecies communication signal, exogenous indole inhibits the persister cell waking in *P. aeruginosa* (35). It also reduces growth and motility but increases biofilm formation and enhances antibiotic tolerance of *Agrobacterium tumefaciens*, which was found not to synthesize indole (36).

In this study, we showed that indole produced by AbiS in *A. baumannii* repressed the QS system and type III secretion system (T3SS) of *P. aeruginosa* and then enhanced the competitive advantages of *A. baumannii* against *P. aeruginosa*. In-frame deletion of *abiS* disrupted biofilm formation, motility and virulence in *A. baumannii*. Furthermore, we also found that indole controls the AHL QS system in *A. baumannii*. As *A. baumannii* and *P. aeruginosa* usually coexist in the human lung, our data suggest that in addition to the important role of indole in the physiology and pathogenesis of *A. baumannii*, it also engages in interspecies competition, the importance of which in survival and infection is evident.

## RESULTS

**The ethyl acetate extract of *A. baumannii* interferes with the QS system and T3SS of *P. aeruginosa*.** As *A. baumannii* and *P. aeruginosa* usually share the same niche in the human lung, we then investigated whether there is competition or a collaborative relationship between the two pathogens. We first extracted low-molecular-weight compounds from a 1 L liquid culture of *A. baumannii* using ethyl acetate and then concentrated the extract to 1 mL with methanol. The results showed that the 1% ethyl acetate extract exerted an inhibitory effect on the motility and cytotoxicity of *P. aeruginosa* (Fig. 1a and b). As *P. aeruginosa* has evolved both QS systems and T3SS for the regulation of important biological functions and pathogenicity (10, 34, 37), we continued to test the effect of the extract and found that exogenous addition of the ethyl acetate extract of *A. baumannii* could reduce the transcriptional expression of both the master regulator-encoding gene of the T3SS and the regulator-encoding genes of QS systems (Fig. 1c and d), that is, *lasR*, *rhlR* and *pqsR*, but did not obviously affect the bacterial growth of *P. aeruginosa* (Fig. S1 in the supplemental material), suggesting that *A. baumannii* produces some compounds that interfere with the physiology and pathogenicity of *P. aeruginosa*.

**The major active components of *A. baumannii* are indole and its derivatives.** To identify the active components of *A. baumannii* that interfere with the physiology and pathogenesis of *P. aeruginosa*, we isolated and purified the active fractions from 140 L

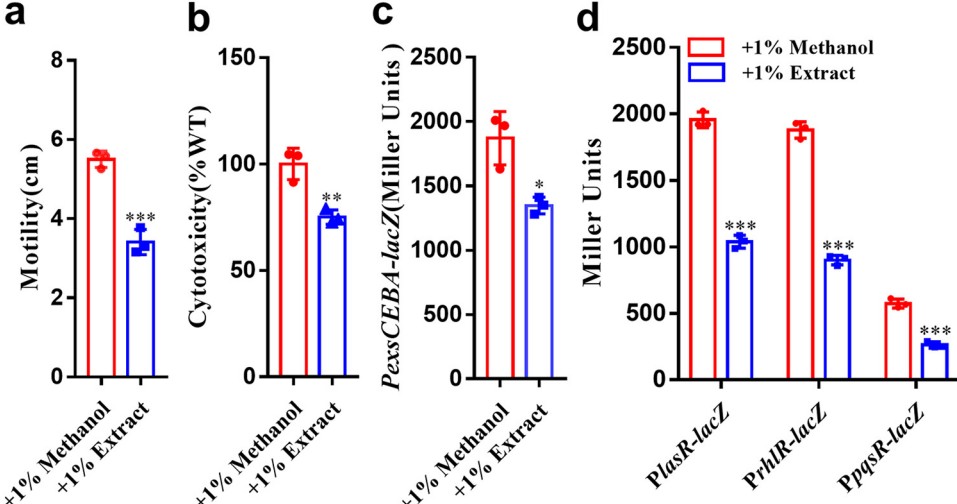

**FIG 1** Effects of ethyl acetate extract of *A. baumannii* on *P. aeruginosa*. Effect of the 1% *A. baumannii* extract on the motility (a) and cytotoxicity (b) of *P. aeruginosa* PAO1, the transcriptional expression of both the master regulator-encoding gene of the T3SS (c) and the regulator-encoding gene of the QS systems (d). Extract was dissolved in methanol, and the same volume of methanol used as the solvent for the compounds was used as a control. The data are the means ± standard deviations of three independent experiments. The significance of result a, b, c was determined by unpaired *t* test. The significance of result d was determined by two-way ANOVA (*, $P < 0.05$; **, $P < 0.01$; ***, $P < 0.001$; ns = no significance).

*A. baumannii* ATCC17978 culture supernatants using high-performance liquid chromatography (HPLC). There was a total of four peaks with active components, which were collected and purified for determination of the chemical structures (Fig. S2a in the supplemental material). Among these peaks, approximately 6.2 mg of the purified compound of peak 4 was purified. ESI-MS spectrometry analysis of the active compound revealed a molecular ion $[M+H]^+$ with an *m/z* ratio of 118.2 (Fig. S2b), consistent with the molecular formula of $C_8H_7N$. There were six protons in the aromatic region and one active hydrogen in the $^1H$ NMR spectrum (Table 1). The $^{13}C$ NMR data of the compound showed the presence of eight aromatic carbons (Table 1). The results are consistent with the literature (38), which indicated that the active compound was indole (Fig. S2b in the supplemental material). In addition, three indole derivatives were also isolated from the active ingredients and characterized: peak 1 was identified as indole-3-acetic acid (IAA), peak 2 was identified as 2-(2,2-di [1*H*-indole-3-yl] ethyl) aniline (di-IEA), and peak 3 was identified as methyl 2-(1*H*-indole-3-yl) acetate (MIA). The electrospray ionization-tandem mass spectrometry (ESI-MS) spectrometry analysis results of the three active compounds are shown in Fig. S2c, 2d, and 2e. A total of 100 $\mu$M indole and its derivatives showed obvious inhibitory activity on the motility and cytotoxicity of *P. aeruginosa* PAO1 (Fig. 2). We also measured the effect of indole and its derivatives on the growth rate of *P. aeruginosa* PAO1 and found that exogenous addition of 100 $\mu$M indole and its derivatives had little influence on the growth rate of *P. aeruginosa* PAO1 in both LB and MP culture media (Fig. S3 in the supplemental material).

**Indole and its derivatives interfere with the T3SS and QS systems of *P. aeruginosa*.** As indole and its derivatives inhibited the motility and cytotoxicity of *P. aeruginosa*, we continued to investigate the mechanisms by which these compounds affect *P. aeruginosa*. We first evaluated the effect of indole and its derivatives on the T3SS system of *P. aeruginosa*. The real-time quantitative reverse transcription-PCR (RT-qPCR) and P*exsCEBA-lacZ* fusion reporter analyses showed that the addition of 100 $\mu$M indole and IAA decreased the transcriptional expression levels of *exsCEBA* (Fig. 3). However, di-IEA and MIA had no inhibitory activities (Fig. 3).

Then, we evaluated the effect of indole and its derivatives on the QS systems of *P. aeruginosa*. The results of fluorescence RT-qPCR and promoter-*lacZ* fusion reporter assays showed that the expression levels of *lasI*, *rhlI*, *pqsA*, *lasR*, *rhlR* and *pqsR* were significantly reduced with the addition of indole (Fig. S4a and c in the supplemental material). Consistent with these results, the production of C4-HSL, 3-oxo-C12-HSL and PQS

**TABLE 1** $^1$H NMR (400 MHz) and $^{13}$C NMR (101 MHz) data of indole and its derivatives

| No. | Indole$^a$ $\delta_C$ | $\delta_H$ | di-IEA$^a$ $\delta_C$ | $\delta_H$ | MIA$^a$ $\delta_C$ | $\delta_H$ | IAA$^b$ $\delta_C$ | $\delta_H$ |
|---|---|---|---|---|---|---|---|---|
| NH-1 | | 7.76 | | 7.86 | | 8.11 | | |
| 2 | 124.3 | 7.15 | 122.0 | 7.12 | 123.3 | 7.12 | 124.6 | 7.13 |
| 3 | 102.5 | 6.54 | 119.7 | | 108.3 | | 108.9 | |
| 3a | 127.9 | | 127.0 | | 127.3 | | 128.6 | |
| 4 | 119.9 | 7.67 | 119.2 | 6.91–7.01 | 118.9 | 7.60 | 119.4 | 7.53 |
| 5 | 120.8 | 7.03 | 119.8 | 7.47 | 119.7 | 7.04 | 119.8 | 7.00 |
| 6 | 122.0 | 7.20 | 121.9 | 6.91–7.01 | 122.2 | 7.18 | 122.4 | 7.08 |
| 7 | 111.2 | 7.27 | 111.2 | 7.28 | 111.4 | 7.27 | 112.2 | 7.32 |
| 7a | 135.8 | | 136.7 | | 136.2 | | 138.0 | |
| 8 | | | 34.5 | 4.84 | 31.2 | 3.77 | 31.9 | 3.71 |
| 9 | | | 37.2 | 3.39 | 172.8 | | 176.4 | |
| 10 | | | 126.1 | | 52.1 | 3.68 | | |
| 11 | | | 127.0 | 6.91–7.01 | | | | |
| 12 | | | 115.8 | 6.53 | | | | |
| 13 | | | 130.4 | 6.91–7.01 | | | | |
| 14 | | | 118.8 | 6.61 | | | | |
| 15 | | | 144.8 | | | | | |
| NH-1′ | | | | 7.86 | | | | |
| 2′ | | | 122.0 | 7.12 | | | | |
| 3′ | | | 119.7 | | | | | |
| 3′a | | | 127.0 | | | | | |
| 4′ | | | 119.2 | 6.91–7.01 | | | | |
| 5′ | | | 119.8 | 7.47 | | | | |
| 6′ | | | 121.9 | 6.91–7.01 | | | | |
| 7′ | | | 111.2 | 7.28 | | | | |
| 7′a | | | 136.7 | | | | | |

$^a$The solvent was CDCl$_3$.
$^b$The solvent was CD$_3$OD.

was decreased significantly in the presence of indole (Fig. S4b). The three derivatives showed obvious inhibitory activities on the QS system, except exogenous addition of 100 $\mu$M IAA, which exhibited no detectable effect on the transcriptional expression levels of *lasR* and *pqsR* (Fig. S4d–f).

**Biosynthesis of indole is performed by AbiS in *A. baumannii*.** Previous studies have confirmed the important role of indole in cell–cell communication. To further study the functions of indole in *A. baumannii*, we first looked for the enzyme involved in the biosynthesis of indole and its derivatives. When we selected TnaA from *E. coli* for

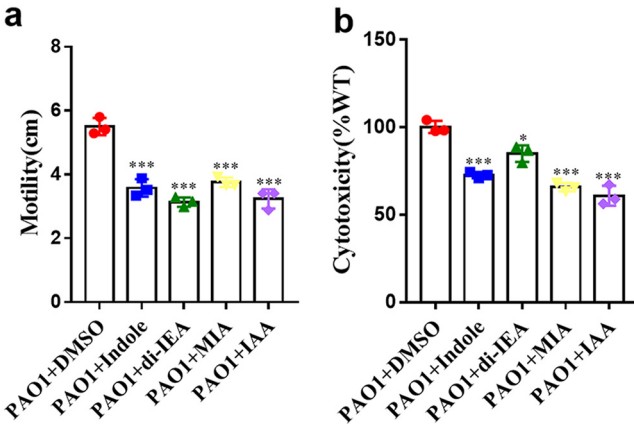

**FIG 2** Effects of indole and indole derivatives on the biological functions of *P. aeruginosa*. Exogenous addition of 100 $\mu$M indole, 100 $\mu$M di-IEA, 100 $\mu$M MIA and 100 $\mu$M IAA on the motility (a) and cytotoxicity (b) of *P. aeruginosa* PAO1. Compounds was dissolved in DMSO, and the same volume of DMSO used as the solvent for the compounds was used as a control. The data are the means ± standard deviations of three independent experiments. The significance was determined by one-way ANOVA (*, $P < 0.05$; **, $P < 0.01$; ***, $P < 0.001$; ns = no significance).

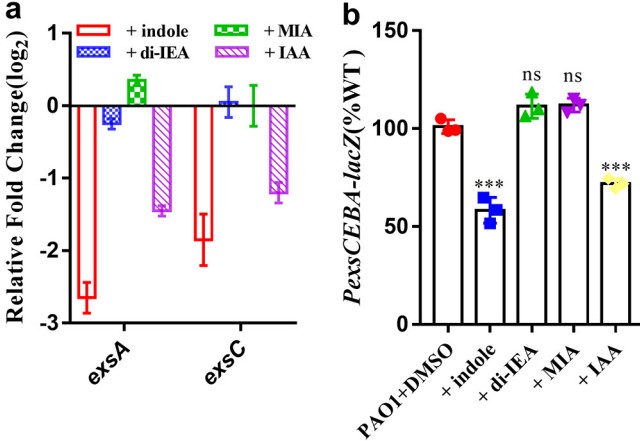

**FIG 3** Effects of indole and indole derivatives on the T3SS system of *P. aeruginosa*. (a) The effects of 100 $\mu$M indole, 100 $\mu$M di-IEA, 100 $\mu$M MIA, and 100 $\mu$M IAA on the expression of the master regulator-encoding gene of the T3SS were evaluated by RT-qPCR (OD$_{600}$ = 1.0). (b) Inhibitory effect of 100 $\mu$M indole, 100 $\mu$M di-IEA, 100 $\mu$M MIA, and 100 $\mu$M IAA on the expression of the T3SS system, as determined by using P*exsCEBA-lacZ* transcriptional fusion reporter strains (OD$_{600}$ = 3.0). The $\beta$-galactosidase activity of P*exsCEBA-lacZ* in the *P. aeruginosa* PAO1 wild-type strain was arbitrarily defined as 100% and used to normalize the $\beta$-galactosidase activity of P*exsCEBA-lacZ* in the *P. aeruginosa* PAO1 strains supplemented with indole and its derivatives. Compounds was dissolved in DMSO, and the same volume of DMSO used as the solvent for the compounds was used as a control. The data are the means ± standard deviations of three independent experiments. The significance was determined by one-way ANOVA (*, $P < 0.05$; **, $P < 0.01$; ***, $P < 0.001$; ns = no significance).

homology analysis in *A. baumannii*, we found no homologous protein in *A. baumannii*. But we noticed that EstC in *Burkholderia cenocepacia* was noted to be a methyl indole-3-acetate methyltransferase. Given the fact that EstC is also an esterase in some other bacteria, so we supposed that EstC is possibly involved in the production of indole. Homologs of *estC* were then searched for in the genome sequence of *A. baumannii* ATCC17978, and A1S_3160 was identified as one such homolog by using the National Center for Biotechnology Information Basic Local Alignment Search Tool (BLAST) program (39). The A1S_3160 protein has an *alpha/beta* hydrolase fold (Fig. 4a), and mutation of this protein completely abolished indole production (Fig. S5) but resulted in only a slight decrease in the growth rate of the bacterial cells in both LB medium and MP medium (Fig. S6). We named this enzyme <u>*A*</u>cinetobacter <u>*b*</u>aumannii <u>*i*</u>ndole <u>*s*</u>ynthase (AbiS).

To investigate whether the biosynthesis of indole is related to cell density, we measured the time course of indole production by determining indole concentrations at various times (Fig. 4b). The yield of indole was quantitatively calculated by the correlation equation determined by the standard curve (Fig. S7 in the supplemental material). BLAST searches revealed that the AbiS homologs are widely conserved in many other bacterial genera, including *Acinetobacter*, *Marinobacter*, *Klebsiella*, and *Salmonella* (Table S1).

**The indole biosynthetic pathway in *A. baumannii* is different from that in *E. coli*.** A previous study indicated that TnaA can reversibly convert tryptophan to indole, pyruvate and ammonia (19). To date, this is the sole indole production pathway identified in bacteria (15). BLAST searches revealed that the TnaA homologs were also conserved in many bacterial genera (Table S2 in the supplemental material). Moreover, it was found that some bacterial species have both TnaA and AbiS homologs (Table S3). To determine whether the biosynthetic pathway of indole in *A. baumannii* is the same as or different from that in *E. coli*, we knocked out *tnaA* in *E. coli* K12 and overexpressed AbiS and TnaA in the *E. coli tnaA* mutant, respectively. Then the strains mentioned above were cultured to an OD$_{600}$ of 1.2 in both 0.5 × LB medium and MP+ medium. It was reported that LB contains 0.5–0.6 mM tryptophan (40), so we then cultured these strains in 0.5 × LB firstly. The indole yield of *E. coli* wild-type strain was measured as 208.98 $\mu$M (Fig. 5a). It was found that deletion of *tnaA* completely abolished the production of indole, which was restored to the wild-type strain level in the complemented strain (Fig. 5a). Intriguingly, in *trans*

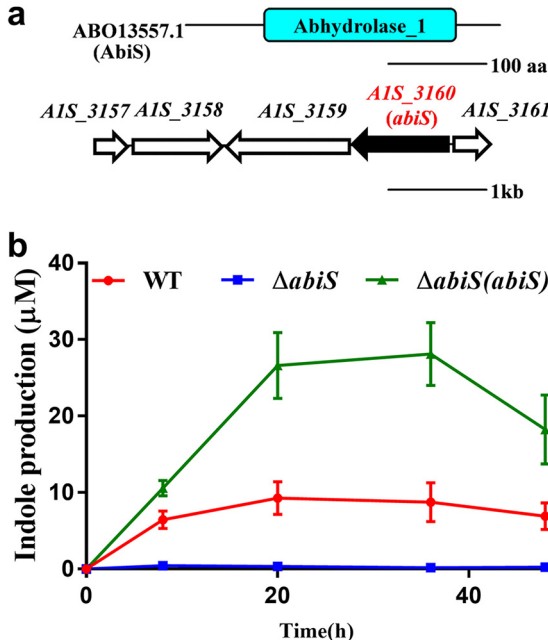

**FIG 4** The biosynthesis of indole is performed by AbiS in *A. baumannii*. (a) Domain structure analysis of AbiS (top). Genomic organization of the *abiS* region in *A. baumannii* ATCC17978 (bottom). (b) Time-course analysis of indole accumulation in the *A. baumannii* wild-type strain, the Δ*abiS* deletion mutant strain and the Δ*abiS*(*abiS*) complemented strain. The data are the means ± standard deviations of three independent experiments.

expression of AbiS in the *tnaA* deletion mutant only slightly rescued the indole production to 21.76 $\mu$M (Fig. 5a). We also found that exogenous addition of 300 $\mu$M tryptophan can significantly increase indole production in *E. coli* to 477.97 $\mu$M. Meanwhile, the addition of 300 $\mu$M tryptophan had no effect on the indole yield of Δ*tnaA*(*abiS*) strain (Fig. 5a). To further confirm the different pathways conducted by AbiS and TnaA, we cultured the *E. coli* wild-type, *tnaA* deletion mutant, *tnaA*(*tnaA*) complement, and *tnaA*(*abiS*) strains in the tryptophan-deficient MP+ medium. It was found that both the indole yields of *E. coli* wild-type and complement strains were remarkable lower than those in 0.5 × LB, while the Δ*tnaA*(*abiS*) strain produced the similar amount of indole in both tryptophan-rich and tryptophan-deficient media (Fig. 5b), suggesting that tryptophan is not a precursor for the biosynthesis of indole by AbiS.

**Deletion of *abiS* impairs biological functions and pathogenicity in *A. baumannii*.** It was determined that indole and its derivatives interfere with the QS systems and T3SS of *P. aeruginosa* through interspecies communication. We continued to study whether indole plays an important role in the regulation of biological functions and pathogenicity in *A. baumannii*. As deletion of *abiS* completely abolished the production of indole, we tested the phenotypic changes in biofilm formation, motility and cytotoxicity in an in-frame deletion mutant of *abiS*. Deletion of *abiS* decreased biofilm formation and motility by 25.49% and 61.91%, respectively (Fig. 6a and b), while it attenuated cytotoxicity by 52.66% when A549 cells were incubated with the *A. baumannii* strains at 8 h postinoculation (Fig. 6c).

Interestingly, both the in *trans* expression of *abiS* and the addition of exogenous indole restored these phenotypes of the *abiS* deletion mutant to the wild-type strain level (Fig. 6). Further investigation was conducted to analyze the symptoms of mouse lungs infected by the mutant strains because the lung is the most important niche for *A. baumannii*. At 7 d postinfection, the macroscopic pathological findings indicated that, compared with the normal group (Fig. S8a in the supplemental material), a significant reduction in the number of alveoli and obvious injury of the epithelial cell lining of bronchioles were observed in the groups infected by the wild-type (Fig. S8b) and complemented (Fig. S8d) strains. In addition, some alveolar walls were severely damaged or broken, and the microvasculature was congested and bled when the mice

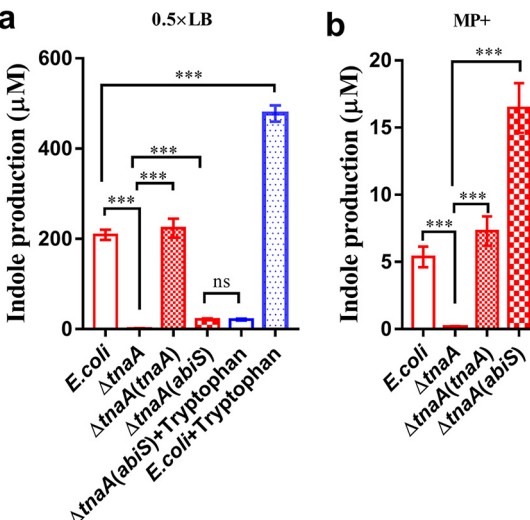

**FIG 5** Analysis of indole biosynthesis performed by AbiS. (a) Analysis of the production of indole in the *E. coli* wild-type, Δ*tnaA*, Δ*tnaA(tnaA)* and Δ*tnaA(abiS)* in 0.5 × LB medium, and in the Δ*tnaA(abiS)* and *E. coli* wild-type strains in 0.5 × LB medium supplemented with 300 μM tryptophan. (b) Analysis of the production of indole in the *E. coli* wild-type, Δ*tnaA*, Δ*tnaA(tnaA)*, and Δ*tnaA(abiS)* strains in MP+ medium. The data are the means ± standard deviations of three independent experiments. The significance was determined by one-way ANOVA (*, $P < 0.05$; **, $P < 0.01$; ***, $P < 0.001$; ns = no significance).

were infected with the wild-type and complemented strains, while only mild injury was observed when the mice were infected with the *abiS* mutant strain (Fig. S8c).

**Indole controls the AHL QS system and the expression of various genes in *A. baumannii*.** The QS system of *A. baumannii* consists of an AHL synthase (AbaI) and a transcription regulator (AbaR), which is activated upon binding to AHL. N-(3-hydroxy-dodecanoyl)-L-homoserine lactone (3-OH-C12-HSL) is the major AHL signal found in *A. baumannii* ATCC17978 (29). The RT-qPCR analysis and the promoter-*lacZ* fusion reporter assay results showed that mutation of *abiS* caused a decrease in the expression level of *abaI* (Fig. 7a and b). We also found that the production of 3-OH-C12-HSL was reduced in the *abiS* mutant strain by 54.52% (Fig. 7c). These results demonstrated that indole positively regulated the AHL QS system in *A. baumannii*.

To determine the comprehensive regulatory roles of indole in controlling bacterial physiology and pathogenicity, we analyzed and compared the transcriptomic profiles of the wild-type strain and the Δ*abiS* mutant strain by using RNA sequencing. Differential gene expression analysis showed that 85 genes were upregulated and 140 genes were downregulated (log$_2$ fold change ≥1.0) in the Δ*abiS* mutant strain compared with the wild-type strain. These differentially expressed genes were associated with a range of biological functions, including transport and metabolism, cell envelope biogenesis, cell motility and secretion, coenzyme metabolism, defense mechanisms, lipid metabolism, signal transduction mechanisms, transcription and translation ribosomal structure and biogenesis (Fig. S9 and Table S4 in the supplemental material). Further analysis showed that some virulence-related genes, such as *epsA* (A1S_0051), *pgaA* (A1S_2162), and *plD* (A1S_2099), were also downregulated in the *abiS* deletion mutant strain (Fig. S10).

**Null mutation of AbiS decreases the competitive capability of *A. baumannii* against *P. aeruginosa* in the mixed culture.** As indole interferes with the QS systems and T3SS of *P. aeruginosa* and plays a vital role in the regulation of the physiology and pathogenicity of *A. baumannii*, we investigated whether there exist any competitive interactions between these two bacterial species, and if so, whether indole plays a role in interspecies competition. To this end, we cocultured the *P. aeruginosa* PAO1 strain with fluorescence labeling in the absence or presence of *A. baumannii* strains. The results showed that the GFP mean fluorescence intensity (MFI) of *P. aeruginosa* was $6.7 × 10^6$ when it was grown alone (Fig. 8a). The value decreased to $2.83 × 10^6$ when *P. aeruginosa* was cocultured with the *A. baumannii* wild-type strain. The GFP MFI of *P. aeruginosa*

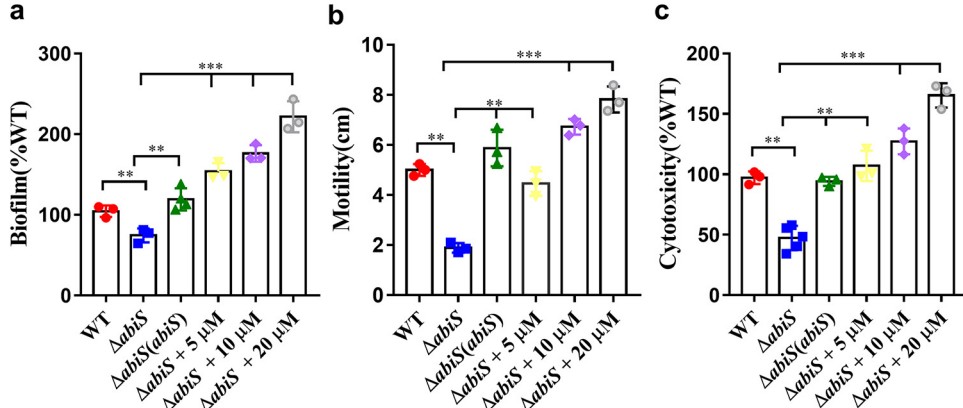

**FIG 6** Effects of *abiS* and indole on the biological functions of *A. baumannii*. The virulence-related phenotypes of biofilm formation (a), motility (b) and cytotoxicity (c) in the *A. baumannii* wild-type, *abiS* mutant, *abiS* complemented strains and *abiS* mutant with addition of different concentrations of indole were examined. Compounds was dissolved in DMSO, and the same volume of DMSO used as the solvent for the compounds was used as a control. The data are the means $\pm$ standard deviations of three independent experiments. The significance was determined by one-way ANOVA (*, $P < 0.05$; **, $P < 0.01$; ***, $P < 0.001$; ns = no significance).

cocultured with the deletion mutant $\Delta abiS$ was approximately 148% that of *P. aeruginosa* cocultured with the *A. baumannii* wild-type strain at 48 h postinoculation (Fig. 8a). The results showed that deletion of *abiS* caused a drastic reduction in the competitive capability of *A. baumannii* against *P. aeruginosa*. We also found that deletion of *abiS* caused an approximately 37% reduction in mCherry fluorescence intensity when the cells were cocultured with *P. aeruginosa* at 48 h postinoculation compared with the *A. baumannii* wild-type strain (Fig. 8b). These results suggest that the *abiS* gene-encoded products or functions might play a key role in enhancing the competitive fitness of *A. baumannii* in the microbial community.

## DISCUSSION

In microbial communities, many bacteria use small diffusible signal molecules to sense local environmental conditions and coordinate multicellular behavior (41). In this study, we found that indole and its derivatives play a role in microbial ecology and chose indole as the representative metabolite, as it has been revealed to be a versatile interspecies and interkingdom signaling molecule in prokaryotic and eukaryotic com-

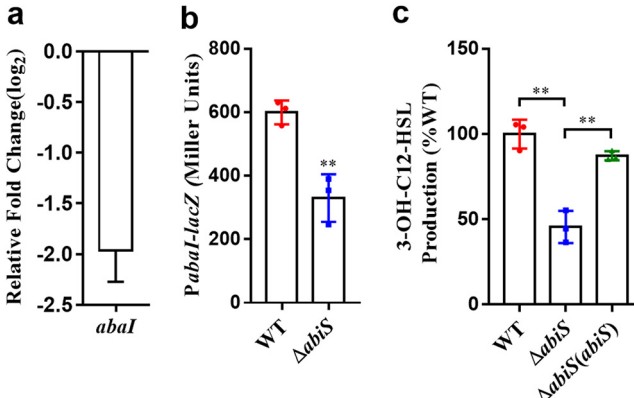

**FIG 7** Effects of *abiS* on the AHL QS system of *A. baumannii*. The expression of the AHL signal synthase-encoding gene was evaluated by RT-qPCR (a) and by assessing the $\beta$-galactosidase activity of the promoter-*lacZ* transcriptional fusions in the wild-type and *abiS* mutant strains (OD$_{600}$ = 3.0) (b). The production of the AHL signal in the *A. baumannii* wild-type strain was arbitrarily defined as 100% and used to normalize the amount of that signal in the *abiS* mutant and complemented strains (c). The data are the means $\pm$ standard deviations of three independent experiments. The significance of result b was determined by *unpaired t test*. The significance of result c was determined by one-way ANOVA (*, $P < 0.05$; **, $P < 0.01$; ***, $P < 0.001$; ns = no significance).

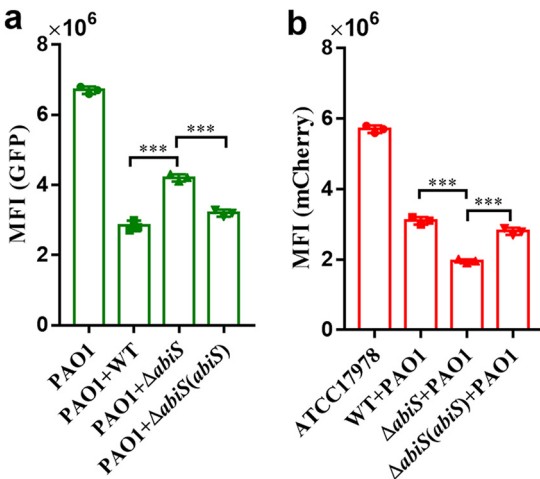

**FIG 8** Null mutation of AbiS decreases the competitive capability of *A. baumannii* against *P. aeruginosa* in the mixed culture. The green fluorescent protein expression vector was carried by the *P. aeruginosa* PAO1 wild-type strain. The ATCC17978 wild-type, Δ*abiS* and Δ*abiS*(*abiS*) strains carried the mCherry fluorescent protein expression vector. The ATCC17978 wild-type, Δ*abiS* and Δ*abiS*(*abiS*) strains were cocultured with *P. aeruginosa* PAO1 at a ratio of 4:1 (vol/vol) at $OD_{600}$ = 0.1. (a) Green mean fluorescence intensity of the *P. aeruginosa* PAO1 wild-type strain carrying the green fluorescent protein expression vector. (b) Red mean fluorescence intensity of the ATCC17978 WT, Δ*abiS* and Δ*abiS*(*abiS*) strains carrying the mCherry fluorescent protein expression vector. The data are the means ± standard deviations of three independent experiments. The significance was determined by one-way ANOVA (*, $P < 0.05$; **, $P < 0.01$; ***, $P < 0.001$; ns = no significance).

munities (15, 16, 36). Interestingly, we found that indole and its derivatives from *A. baumannii* interferes with the T3SS and QS systems of *P. aeruginosa* (Fig. 3, Fig. S4 in the supplemental material), and indole promotes the competitive advantages of *A. baumannii* against *P. aeruginosa* (Fig. 8). As *P. aeruginosa* and *A. baumannii* are human pathogens that coexist in the lungs, our data not only enhance the important role of indole in interspecies communication but also reveal complicated interactions, including competition and collaboration, among human pathogens.

Previous studies showed that indole has diverse signaling roles, including in modulation of biofilm formation, virulence, stress responses and increase of epithelial-cell tight-junction resistance and decrease of inflammation indicators (17, 20, 32, 33, 42–50). In this study, in addition to playing an important role as an interspecies communication signal, indole also contributed to the physiological function of *A. baumannii*. Deletion of indole synthase impaired biofilm formation, motility and virulence of *A. baumannii* (Fig. 6). Previous studies showed that there is an AHL-based QS system, AbaI/AbaR, playing a vital role in the regulation of biofilm formation, motility and virulence in *A. baumannii* (27–30). As the indole signal also controls these AHL-regulated functions in *A. baumannii*, we then investigated whether there is a relationship between the two kinds of signaling systems. It was observed that both the expression of *abaI* and the production of 3-OH-C12-HSL were significantly reduced in the indole-null mutant (Fig. 7), suggesting a complex hierarchy of signaling systems in *A. baumannii* where indole is upstream of the AHL QS signaling system. However, the detailed regulatory mechanism of indole in *A. baumannii* still needs further investigation.

It has been previously reported that the indole synthase tryptophanase (TnaA) exists in *E. coli*. To date, this is the only indole synthase identified in bacteria (15); this enzyme reversibly converts tryptophan to indole, pyruvate and ammonia (19). In contrast to TnaA, AbiS contains an alpha/beta hydrolase fold. Indole production by TnaA in *E. coli* is determined by exogenous tryptophan concentrations (40).To determine the biosynthesis pathways of indole performed by AbiS, we studied the following aspects. First, overexpression of AbiS in *E. coli* K12 *tnaA* mutant strain rescued the production of indole in both 0.5 × LB and MP+ culture medium (Fig. 5), suggesting that the same precursors that could be converted to indole by AbiS were present in *E. coli* as in

*A. baumannii*. Second, we found that exogenous addition of 300 $\mu$M tryptophan could not affect the production of indole in the $\Delta$ *tnaA*(*abiS*) strain but significantly increased the production of indole in the *E. coli* wild-type strain in 0.5 $\times$ LB (Fig. 5a), suggesting that tryptophan is not a precursor for the biosynthesis of indole by AbiS. Moreover, the $\Delta tnaA$(*abiS*) strain produced the similar amount of indole in both tryptophan-rich and tryptophan-deficient media (Fig. 5b). Together, these findings suggested that the indole biosynthetic pathway performed by AbiS in *A. baumannii* is different from that in *E. coli*.

In addition, a BLAST search with the homologs of the two different indole synthases, AbiS and TnaA, revealed that their homologs are highly conserved in many bacterial species (Tables S1 and S2 in the supplemental material). Homologs of AbiS were found in *Acidovorax*, *Acinetobacter*, *Alcanivorax*, *Marinobacter*, and *Noviherbaspirillum* (Table S3), while homologs of TnaA were found in *Acinetobacter*, *Aeromonas*, *Citrobacter*, *Edwardsiella*, and *Haemophilus* (Table S2). More interestingly, some bacterial genera, including *Acinetobacter*, *Aeromonas*, *Chromobacterium*, *Vibrio*, and *Grimontia*, have homologs of both AbiS and TnaA (Table S3), suggesting that they may have two different biosynthetic pathways to produce indole. In general, our data suggest that the indole/AbiS system is a unique signaling mechanism that is present in various bacterial species. Our findings will trigger further investigation of the roles and mechanisms of this signaling system in diverse bacterial genera.

## MATERIALS AND METHODS

**Ethics statement.** This study was approved by the ethics committee of School of Pharmaceutical Sciences (Shenzhen), Sun Yat-sen University (SYSU-20200404), and all participants gave informed consent.

**Bacterial strains and growth conditions.** The bacterial strains used in this study are listed in Table S5. All the bacterial strains and plasmids used in this study have been sequenced. *A. baumannii*, *P. aeruginosa*, and *E. coli* strains were cultured in Luria-Bertani (LB) medium (10 g/L tryptone, 5 g/L yeast extract, 5 g/L NaCl; pH 7.0–7.5) or LB agar (LB medium containing 15 g/L agar) at 37°C. MP minimal medium (1 L): $FeSO_4\cdot7H_2O$, $1.25 \times 10^{-4}$ g; $(NH_4)_2SO_4$, 0.5 g; $MgSO_4\cdot7H_2O$, 0.05 g; $KH_2PO_4$, 3.4 g. The pH was adjusted to 7.0. MP+ medium: MP minimal medium added with 1% of a 14 amino acid mixture (14 amino acid mixture contained alanine, arginine, aspartic acid, cysteine, glutamic acid, glycine, histidine, isoleucine, leucine, lysine, methionine, serine, threonine, and valine). The antibiotics were added to the medium according to the experimental needs, and the following antibiotics were used in this work: gentamicin (50 $\mu$g/mL), tetracycline (10 $\mu$g/mL), apramycin (100 $\mu$g/mL), kanamycin (50 $\mu$g/mL) or ampicillin (100 $\mu$g/mL). Indole, di-IEA, MIA, IAA (HPLC $\geq$ 99%) were dissolved in DMSO to a final concentration of 100 mM, and the solutions were added to the medium in the experiments, respectively.

**Purification and structural analysis.** *A. baumannii* ATCC17978 cells were cultured overnight in LB medium, and the supernatant was extracted with an equal volume of ethyl acetate. After the ethyl acetate was concentrated by evaporation, the extract was dissolved in methanol and subjected to HPLC analysis on a $C_{18}$ reverse-phase column (Atlantis T3 Column, 5 $\mu$m, 4.6 mm $\times$ 250 mm) and then eluted with acetonitrile-water (from 10:90 to 70:30 vol/vol) at a flow rate of 1 mL/min. The active fractions were detected, concentrated, and purified by HPLC using a semipreparative $C_{18}$ reverse-phase column. Peaks were monitored using an UV detector at 210 nm.

The $^{1}$H and $^{13}$C nuclear magnetic resonance (NMR) spectra were recorded on an AVANCE III HD 400 (Temperature 298.0 K, Bruker, Billerica, MA, USA) operating at 400 MHz for $^{1}$H or 101 MHz for $^{13}$C. UHPLC-ESI-MS/MS was performed in a Shimadzu LC-30A UHPLC system (Shimadzu Corporation, Kyoto, Japan) with a Waters $C_{18}$ column (1.8 $\mu$m, 150 $\times$ 2.1 mm) and a Shimadzu 8060 QQQ-MS mass spectrometer with an ESI source interface (Shimadzu Corporation, Kyoto, Japan).

**Construction of in-frame deletion mutants.** Gene knockout was achieved by a highly efficient CRISPR-Cas9-based genome engineering platform in *A. baumannii* as previously described (51). The CRISPR-Cas9 system is a genome editing tool developed in recent years with strong DNA site-directed cleavage capability. By coupling the RecAb recombination system and the CRISPR-Cas9 genome cleavage system, a two-plasmid genome-editing system, pCasAb/pSGAb, was used for gene deletion. The 80-nt ssDNA was used for double-strand break (DSB) repair in *A. baumannii* ATCC 17978. Meanwhile, the target gene was integrated into the chromosome to obtain the complemented strains by using pWH1266. The resulting constructs were introduced into *A. baumannii* ATCC17978 deletion mutants using electroporation. The primers used for this study are listed in Table S6.

**Construction of reporter strains and measurement of $\beta$-galactosidase activity.** The promoter fragments were inserted upstream of the promoterless *lacZ* gene in pME2-*lacZ*. The *lacZ* fusion constructs were transformed into wild-type and *abiS* deletion mutant strains of *A. baumannii* ATCC17978 by electroporation to obtain the relevant reporter strains. Two hundred $\mu$L of bacterial cultures ($OD_{600}$ = 3.0) were collected to isolate the cells to determine the $\beta$-galactosidase activity as previously described (52). Each experiment was repeated three times in parallel.

**Competition assays in mixed culture.** The green fluorescent protein expression vector was used and transformed into the *P. aeruginosa* PAO1 wild-type strain. The mCherry fluorescent protein expression vector was introduced into the ATCC17978 wild-type strain, $\Delta abiS$ deletion mutant strain and $\Delta abiS$

(*abiS*) complemented strain by electroporation. Then, the *A. baumannii* strains ($OD_{600}$ = 0.1) were cocultured with *P. aeruginosa* ($OD_{600}$ = 0.1) at an initial ratio of 4:1 at 37°C with shaking at 220 rpm for 48 h. Then, the mixed cultures were analyzed by a Spectra Max i3x multifunctional enzyme labeling instrument (Molecular Devices, CA, USA).

**Statistical analysis.** Statistical analyses were performed using Prism 8 software (GraphPad). *Unpaired t test* between two groups, one-way analysis of variance (ANONA) or two-way analysis of variance among multiple groups were used to calculate *P* values. Statistical significance is indicated as follows: ns = no significance; *, $P < 0.05$; **, $P < 0.01$; ***, $P < 0.001$; ****, $P < 0.0001$. All results were calculated from the average of at least three replicates.

**Data availability.** Data supporting the findings of this study are available within the paper and its supplementary information files.

## SUPPLEMENTAL MATERIAL

Supplemental material is available online only.

**SUPPLEMENTAL FILE 1**, PDF file, 1.7 MB.

## ACKNOWLEDGMENTS

This work was financially supported by the National Key Research and Development Program of China (2021YFA0717003), and the Science, Technology and Innovation Commission of Shenzhen Municipality (No. JCYJ20200109142416497).

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
