## [Reviewer comments · Microbiology Spectrum]

Microbiology Spectrum

The cell-cell communication signal indole controls the physiology and interspecies communication of *Acinetobacter baumannii*

Binbin Cui, Xiayu Chen, Quan Guo, Shihao Song, Mingfang Wang, Jingyun Liu, and Yinyue Deng

Corresponding Author(s): Yinyue Deng, Sun Yat-sen University

Review Timeline:

Submission Date:	March 21, 2022
Editorial Decision:	April 11, 2022
Revision Received:	May 19, 2022
Editorial Decision:	May 29, 2022
Revision Received:	June 16, 2022
Accepted:	June 16, 2022

Editor: Beile Gao

Reviewer(s): Disclosure of reviewer identity is with reference to reviewer comments included in decision letter(s). The following individuals involved in review of your submission have agreed to reveal their identity: Jintae Lee (Reviewer #2)

Transaction Report:

DOI: <https://doi.org/10.1128/spectrum.01027-22>

April 11, 2022

Prof. Yinyue Deng
Sun Yat-sen University
Guangzhou 510642
China

Re: Spectrum01027-22 (The cell-cell communication signal indole modulates the physiology and competitive ability of *Acinetobacter baumannii*)

Dear Prof. Yinyue Deng:

Link Not Available

Sincerely,

Beile Gao

Journals Department
Reviewer comments:

Reviewer #1 (Comments for the Author):

The authors indicate that indole increases *A. b.* virulence by increasing AHL synthesis, confirm indole reduces *P. a.* virulence (as indicated by ref. 32), and claim indole is produced by an enzyme different from tryptophanase in *A. b.* The manuscript is well-written. "*" indicates major problems below which need to be addressed. One of the main issues is the lack of solvent (negative controls) throughout; this is critical given the mistakes in the indole literature related to the use of solvents as indole diluents (DMSO here, which has significant toxicity, if not used less than 0.2 vol%). Also, the constructed strains must be validated, and *AbiS* was not shown to synthesize indole (it was shown genetically to influence indole concentrations).

*1. Title and l 27, 41: replace "modulates" with a more informative word; i.e., indicate how indole affects physiology (increases or

decreases virulence etc.). Avoid vague writing.

2. I 33: indicate more specifically how indole "interfered with its QS" for *P. a.* since it is already published that indole reduces *P. a.* virulence (current ref 32). Replace "interfering" with a more informative word.

3. I 39: add ref for first sentence.

4. I 39: delete "It was demonstrated that" here and anywhere else the ms.

5. I 77: add ref doi:10.1111/jam.14434 to indicate that indole reduces *P. a.* persister cell resuscitation since the interaction with *A. b.* is the main point of this work.

*6. Replace ref 19, which has been argued to be plagiarism, with the discovery for indole detection through CpxAR:

Hirakawa H, Inazumi Y, Masaki T, Hirata T, Yamaguchi A. 2004. Indole induces the expression of multidrug exporter genes in *Escherichia coli*. *Mol Microbiol* 55:1113-1126. <https://doi.org/10.1111/j.1365-2958.2004.04449.x>. 2007.

7. I 72: delete "It has been well established".

*8. I 76: replace ref 19 (2nd report and likely plagiarism) with the first report of indole reducing EHEC virulence (2007):

Differential effects of epinephrine, norepinephrine, and indole on *Escherichia coli* O157:H7 chemotaxis, colonization, and gene expression. *Infect Immun* 75:4597- 4607. <https://doi.org/10.1128/IAI.00630-07>.

9. I 78 and I 238: consider adding ref that first showed indole is an inter-species signal by tightening epithelial junctions (www.pnas.org/cgi/doi/10.1073/pnas.0906112107), which has led to many later ones showing indole is important for mood, skin, liver, brain development, aging, etc. Far better to use the primary reference rather than the over-reliance on the indole review articles.

10. I 78: start new paragraph with "In this study..."

*11. Fig. 1abcd lacks ethyl acetate solvent effect (negative controls).

*12. I 118: Fig. S3 lacks DMSO solvent negative controls.

13. I 56 SI: should be "qRT-PCR" and should refer to turbidity, not absorbance (i.e., no OD) for cell growth.

14. I 63 SI: need ref for method.

*15. Fig. S4 lacks solvent negative control. Fix all results in the manuscript.

*16. Fig. 2abc lacks solvent controls.

17. Fig. 2: indole physiological concentrations are around 0.7 mM so please explain the use of 0.1 mM. 2 mM is toxic.

18. I 245: add citation for first use of indole reducing virulence, ref 32, here.

*19. Fig. S5 lacks solvent negative control.

20. I 141: Currently confusing. Explain better why EstC was checked in *A. b.*

*21. L 307: sequence all strains to show constructed correctly.

*22. L 146: Indole detection methods for production in *A. b.* is missing.

*23. AbiS is shown to affect indole concentrations, but it was NOT shown to synthesize indole (could be positive regulator, etc.). Purify the protein (etc.) and demonstrate synthesis or change the claim throughout the text. A new way to synthesize indole is exciting but it is not demonstrated here yet.

*24. Fig. 5 caption does not explain what is being added in the 3 panels. L 184 indicates indole was added but there was no solvent control used, again.

25. I 227: Replace slang "on the other hand" and use better phrasing as it appears this is not what is meant.

26. I 235: should be "behavior".

27. I 245: add first reports of indole and virulence, current ref 32 + <https://doi.org/10.1128/IAI.00630-07> here.

Reviewer #2 (Comments for the Author):

Authors report indole production by AbiS gene from *Acinetobacter baumannii* and its roles in *A. baumannii* and against *Pseudomonas aeruginosa*. While the study is generally interested, there are several serious issues that should be further addressed.

Major comment 1: How much extracellular indole was found in the culture supernatant of *A. baumannii*? If Fig. 3 is correct, the wild type of *A. baumannii* produces only 400 nM (0.4 μ M). It is very small amount compared to other indole-producing strains of *E. coli* and *Vibrio* strains that often produces more than 400 μ M (1000 times more than this study). If so, other assays with indole addition (50 and 100 μ M) is not physiological relevant. For the study of indole biosynthesis by AbiS in *E. coli*, indole deficient *E. coli* (*tnaA* mutant) should be used, for example, Fig. 4A. Then, authors can clearly conclude that indole biosynthesis pathway by AbiS is different from that by *tnaA*.

Major comment 2: While the identification of AbiS gene in this study is novel, the competition assay with indole against *P. aeruginosa* is not new. A key previous study (Indole Production Promotes *Escherichia coli* Mixed-Culture Growth with *Pseudomonas aeruginosa* by Inhibiting Quorum Signaling, *Applied and Environmental Microbiology*, 2012, 78, <https://doi.org/10.1128/AEM.06396-11>). The previous study already reported that indole from *E. coli* inhibited pyocyanin production and other AHL-regulated virulence factors in *P. aeruginosa*, which is similar to Fig. 2 and Fig. 7. Please compare current results to previous results carefully. Also, it has been reported that indole enhanced the competitive ability against other bacteria, such as *Staphylococcus aureus*, *Candida albicans*, *Paenibacillus alvei*, *Serratia marcescens*, etc, which are worthy to mention.

Major comment 3: Authors insisted that tryptophan is not a precursor for the biosynthesis of indole by AbiS. Then what is the indole precursor? Also, authors said that the indole biosynthetic pathway performed by AbiS in *A. baumannii* is different from that in *E. coli*. Then, how *A. baumannii* can produce indole? Also, it has been mentioned that homologs of AbiS were found in *Acidovorax*, *Acinetobacter*, *Alcanivorax*, *Marinobacter* and *Noviherbaspirillum* (Table S3). Then, can these bacteria produce indole?

Reviewer #3 (Comments for the Author):

This manuscript describes a new indole/AbiS system in *A. baumannii* with multiple signalling effects, which would be interesting to a wide range of audience. However, further information and experimental details are needed to confirm the novelty of this system.

Staff Comments:

Preparing Revision Guidelines

Please return the manuscript within 60 days; if you cannot complete the modification within this time period, please contact me. If you do not wish to modify the manuscript and prefer to submit it to another journal, please notify me of your decision immediately so that the manuscript may be formally withdrawn from consideration by Microbiology Spectrum.

Point-to-point response to reviewers' suggestions

Reviewer comments:

Reviewer #1 (Comments for the Author):

The authors indicate that indole increases *A. b.* virulence by increasing AHL synthesis, confirm indole reduces *P. a.* virulence (as indicated by ref. 32), and claim indole is produced by an enzyme different from tryptophanase in *A. b.* The manuscript is well-written. "*" indicates major problems below which need to be addressed. One of the main issues is the lack of solvent (negative controls) throughout; this is critical given the mistakes in the indole literature related to the use of solvents as indole diluents (DMSO here, which has significant toxicity, if not used less than 0.2 vol%). Also, the constructed strains must be validated, and AbiS was not shown to synthesize indole (it was shown genetically to influence indole concentrations).

Response: Thanks a lot for your good comments and suggestions. In our experiments, we have already added the same amount of solvent used for the compounds as a control, we have added this information in the Figure Legend in the revised manuscript. All constructed strains were validated by DNA sequencing. In addition, we found that AbiS was not able to synthesize indole from tryptophan, but it can synthesize indole from other precursors inside cells.

*1. Title and I 27, 41: replace "modulates" with a more informative word; i.e., indicate how indole affects physiology (increases or decreases virulence etc.). Avoid vague writing.

Response: We have changed "modulates" to "controls" and "increases" as suggested.

2. I 33: indicate more specifically how indole "interfered with its QS" for *P. a.* since it is already published that indole reduces *P. a.* virulence (current ref 32). Replace "interfering" with a more informative word.

Response: (Line 33) We have changed this sentence to "Moreover, we revealed that indole from *A. baumannii* reduced the competitive fitness of *Pseudomonas aeruginosa* by inhibiting its QS systems and type III secretion system (T3SS)."

3. I 39: add ref for first sentence.

Response: There is no citation allowed in “Importance” section, but we have cited the relevant reference in the “Introduction” section.

4. I 39: delete "It was demonstrated that" here and anywhere else the ms.

Response: We have deleted it as suggested.

5. I 77: add ref doi:10.1111/jam.14434 to indicate that indole reduces P. a. persister cell resuscitation since the interaction with A. b. is the main point of this work.

Response: We have added this paper as Reference 35 (Zhang W, Yamasaki R, Song S, Wood TK (2019) Interkingdom signal indole inhibits *Pseudomonas aeruginosa* persister cell waking. *J Appl Microbiol* 127:1768-1775) as suggested.

*6. Replace ref 19, which has been argued to be plagiarism, with the discovery for indole detection through CpxAR:

Hirakawa H, Inazumi Y, Masaki T, Hirata T, Yamaguchi A. 2004. Indole induces the expression of multidrug exporter genes in *Escherichia coli*. *Mol Microbiol* 55:1113-1126. <https://doi.org/10.1111/j.1365-2958.2004.04449.x>. 2007.

Response: We have replaced the previous Reference 19 with the above-mentioned reference as Reference 20 as suggested.

7. I 72: delete "It has been well established".

Response: We have deleted it as suggested.

*8. I 76: replace ref 19 (2nd report and likely plagiarism) with the first report of indole reducing EHEC virulence (2007):

Response: We have added the paper “Bansal T, Englert D, Lee J, Hegde M, Wood TK, Jayaraman A. 2007. Differential effects of epinephrine, norepinephrine, and indole on *Escherichia coli* O157:H7 chemotaxis, colonization, and gene expression. *Infect Immun* 75:4597-4607” as Reference 17.

9. I 78 and I 238: consider adding ref that first showed indole is an inter-species signal by tightening epithelial junctions (www.pnas.org/cgi/doi/10.1073/pnas.0906112107), which has led to many later ones showing indole is important for mood, skin, liver, brain development, aging, etc. Far better to use the primary reference rather than the over-reliance on the indole review articles.

Response: We have added the mentioned paper as Reference 42.

10. I 78: start new paragraph with "In this study..."

Response: (Line 89) Thanks a lot for your good suggestion, we have revised it as suggested.

*11. Fig. 1abcd lacks ethyl acetate solvent effect (negative controls).

Response: Extract was dissolved in methanol, and the same amount of methanol used as the solvent for the compounds was added in the control, we have added this information in the figure legend.

*12. I 118: Fig. S3 lacks DMSO solvent negative controls.

Response: Compounds were dissolved in DMSO, and the same amount of DMSO used as the solvent for the compounds was added in the control, we have added this information in the figure legend.

13. I 56 SI: should be "qRT-PCR" and should refer to turbidity, not absorbance (i.e., no OD) for cell growth.

Response: It has been changed to "Quantitative RT-PCR analysis" and " 2×10^9 cfu/mL".

14. I 63 SI: need ref for method.

Response: The paper has been added as Reference 3 in SI (Livak KJ, Schmittgen TD. 2001. Analysis of relative gene expression data using real-time quantitative PCR and the 2(-Delta Delta C(T)) Method. Methods 25:402-408).

*15. Fig. S4 lacks solvent negative control. Fix all results in the manuscript.

Response: Compounds were dissolved in DMSO, and the same amount of DMSO used as the solvent for the compounds was added in the control, we have added this information in the figure legend.

*16. Fig. 2abc lacks solvent controls.

Response: Compounds were dissolved in DMSO, and the same amount of DMSO used as the solvent for the compounds was added in the control, we have added this information in the figure legend.

17. Fig. 2: indole physiological concentrations are around 0.7 mM so please explain the use of 0.1 mM. 2 mM is toxic.

Response: We have tested the effects of indole and its derivatives at different concentrations (25 μ M, 50 μ M and 100 μ M), and found that exogenous addition of 0.1 mM showed a strong activity, so we chose 0.1 mM in these experiments.

18. I 245: add citation for first use of indole reducing virulence, ref 32, here.

Response: We have added Reference 17 as suggested.

*19. Fig. S5 lacks solvent negative control.

Response: Compounds were dissolved in DMSO, and the same amount of DMSO used as the solvent for the compounds was added in the control, we have added this information in the figure legend.

20. I 141: Currently confusing. Explain better why EstC was checked in A. b.

Response: (Line 157) EstC is an esterase in some bacteria, it was also identified to show activity on a variety of ethyl esters and ester compounds consisting of substituted phenyl alcohols and short n-chain fatty acids. When we selected TnaA from *E. coli* for homology analysis in *A. baumannii*, we found no homologous protein in *A. baumannii*. But we noticed that EstC in *Burkholderia cenocepacia* was noted to be a methyl indole-3-acetate methyltransferase. Linking

the fact that EstC is also an esterase in some other bacteria, so we hypothesized that EstC is possibly involved in the production of indole. Then we searched homologs of *estC* in the genome sequence of *A. baumannii* ATCC17978, and A1S_3160 was identified and studied in our manuscript.

*21. L 307: sequence all strains to show constructed correctly.

Response: All the constructed strains and plasmids used in this study have been sequenced to ensure the correct constructions.

*22. L 146: Indole detection methods for production in *A. b.* is missing.

Response: Thanks a lot for your suggestion. We have revised the methods and added it in the Supplementary Information as suggested.

*23. AbiS is shown to affect indole concentrations, but it was NOT shown to synthesize indole (could be positive regulator, etc.). Purify the protein (etc.) and demonstrate synthesis or change the claim throughout the text. A new way to synthesize indole is exciting but it is not demonstrated here yet.

Response: We can conclude that AbiS is involved in the biosynthesis of indole in *Acinetobacter baumannii*, but we have not identified its precursor. Till now, we found that AbiS is different from TnaA (Fig 5b, 5c).

*24. Fig. 5 caption does not explain what is being added in the 3 panels. L 184 indicates indole was added but there was no solvent control used, again.

Response: The Figure order has been rearranged. We have modified this caption as suggested (Fig. 6). The solvent control was also added as suggested.

25. l 227: Replace slang "on the other hand" and use better phrasing as it appears this is not what is meant.

Response: (Line 250) It has been changed to: "We also found that deletion of *abiS* caused an approximately 37% reduction in mCherry fluorescence intensity when the cells were cocultured

with PAO1 at 48 h post-inoculation compared with the *A. baumannii* wild-type strain (Fig. 8b).”

26. | 235: should be "behavior".

Response: (Line 258) It has been changed to: “In microbial communities, many bacteria use small diffusible signal molecules to sense local environmental conditions and coordinate multicellular behavior.”

27. | 245: add first reports of indole and virulence, current ref 32 + <https://doi.org/10.1128/IAI.00630-07> here.

Response: Reference has been added in the position as suggested as Reference 17.

Reviewer #2 (Comments for the Author):

Authors report indole production by AbiS gene from *Acinetobacter baumannii* and its roles in *A. baumannii* and against *Pseudomonas aeruginosa*. While the study is generally interested, there are several serious issues that should be further addressed.

Major comment 1: How much extracellular indole was found in the culture supernatant of *A. baumannii*? If Fig. 3 is correct, the wild type of *A. baumannii* produces only 400 nM (0.4 μ M). It is very small amount compared to other indole-producing strains of *E. coli* and *Vibrio* strains that often produces more than 400 μ M (1000 times more than this study). If so, other assays with indole addition (50 and 100 μ M) is not physiological relevant. For the study of indole biosynthesis

by AbiS in *E. coli*, indole deficient *E. coli* (*tnaA* mutant) should be used, for example, Fig. 4A. Then, authors can clearly conclude that indole biosynthesis pathway by AbiS is different from that by *tnaA*.

Response: (Line 178-195) Thanks a lot for your good suggestions. We have modified the methods to measure indole concentration in the supernatant of *A. baumannii* and revised the Fig. 4C. We have also constructed *E. coli tnaA* deletion mutant and revised Fig. 5b, 5c to clearly confirm that the indole biosynthesis pathway performed by AbiS is different from that by *TnaA*.

Major comment 2: While the identification of AbiS gene in this study is novel, the competition

assay with indole against *P. aeruginosa* is not new. A key previous study (Indole Production Promotes *Escherichia coli* Mixed-Culture Growth with *Pseudomonas aeruginosa* by Inhibiting Quorum Signaling, *Applied and Environmental Microbiology*, 2012, 78, <https://doi.org/10.1128/AEM.06396-11>). The previous study already reported that indole from *E. coli* inhibited pyocyanin production and other AHL-regulated virulence factors in *P. aeruginosa*, which is similar to Fig. 2 and Fig. 7. Please compare current results to previous results carefully. Also, it has been reported that indole enhanced the competitive ability against other bacteria, such as *Staphylococcus aureus*, *Candida albicans*, *Paenibacillus alvei*, *Serratia marcescens*, etc, which are worthy to mention.

Response: Thanks a lot for your good suggestion. We have rearranged and moved Fig. 2 to Fig. S4 in the supplementary information. We have also added the mentioned paper (Chu et al., 2012, *AEM*) as Reference 34, the other papers have been cited as References 44-50 as suggested.

Major comment 3: Authors insisted that tryptophan is not a precursor for the biosynthesis of indole by AbiS. Then what is the indole precursor? Also, authors said that the indole biosynthetic pathway performed by AbiS in *A. baumannii* is different from that in *E. coli*. Then, how *A. baumannii* can produce indole? Also, it has been mentioned that homologs of AbiS were found in *Acidovorax*, *Acinetobacter*, *Alcanivorax*, *Marinobacter* and *Noviherbaspirillum* (Table S3). Then, can these bacteria produce indole?

Response: Thanks a lot for your good suggestions. We have found that tryptophan is not a precursor for the biosynthesis of indole performed by AbiS (Fig. 5b, 5c), but we don't know what is the precursor, which is worth to being investigated in the future. We will also measure the production of indole in other bacterial species in the next project.

Reviewer #3 (Comments for the Author):

This manuscript describes a new indole/AbiS system in *A. baumannii* with multiple signalling effects, which would be interesting to a wide range of audience. However, further information and experimental details are needed to confirm the novelty of this system.

The manuscript describes a new indole synthesis pathway in *A. baumannii* via A1S_3160 protein (named AbiS by the authors). Also, the authors report two novel aspects of indole signalling in *A. baumannii*, including (i) physiological effects on biofilm formation, motility and virulence via modulating AHL QS system, and (ii) interspecies competitive effect against the non-indole producer *P. aeruginosa* via repressing its QS system & T3SS.

The manuscript is well written and clearly structured but there are few general and experimental issues that need to be addressed by the authors. One major issue is related to the authors suggestion that the indole biosynthetic pathway performed by AbiS in *A. baumannii* is different from that of TnaA in *E. coli*. The authors suggest that tryptophan is not a precursor for indole production but provide no clear evidence to support their finding. Can indole be produced by *A. baumannii* in a minimal media lacking tryptophan? Can indole be produced in an *E. coli* Δ tnaA (*abiS*) growing in a minimal media lacking tryptophan? If so, this should support the authors' suggested mechanism.

Provided that tryptophan is not the precursor for indole synthesis via AbiS, the authors did not provide details on the alternative precursor?

Response: Thanks a lot for your good suggestions. We have constructed *E. coli tnaA* deletion mutant and grown these strains in 0.5 x LB and a minimal media lacking tryptophan as suggested. As shown in Fig. 5, exogenous addition of tryptophan will not affect the production of indole performed by AbiS, but will significantly induce the indole production performed by TnaA. In addition, Δ tnaA (*abiS*) produces almost the same amount of indole in the 0.5 x LB and tryptophan deficient minimal media, suggesting that tryptophan will not affect the biosynthesis of indole via AbiS.

Other concerns are summarised below.

Lines 61: CpxA has been recently proposed as an indole sensor in *E. coli*. However, two other envelope stress response modules (BaeSR and the phage shock pathway) respond to indole. Therefore, there has been suggestions that the cell membrane might be responsible for perception of indole. (See Raffa and Raivio, 2002, Vega et al, 2012 & Zarkan et al, 2020). The

authors need to highlight that CpxA is not the only potential indole sensor in *E. coli*.

Response: We have modified these sentences as suggested.

Line 117: Why was the concentration of 100 μ M chosen for the motility and cytotoxicity assays? Were other concentrations tested? (same questions for line 120). What is the supernatant concentration of indole in *A. baumannii* LB cultures? The authors should ideally measure the indole supernatant concentration and use that concentration for their assays. The authors need to comment and clarify the physiological relevance of their chosen concentration.

Response: We have tested the effects of indole and its derivatives at different concentrations (25 μ M, 50 μ M and 100 μ M), and found that exogenous addition of 0.1 mM indole showed a strong activity. In addition, we have also modified the methods to measure indole concentration in the supernatant of *A. baumannii* and revised the results in Fig. 4b, which showed that the supernatant concentration of indole culture in LB medium is about 10 μ M. We also found that addition of 10 μ M indole is efficient to restore the defective phenotypes in the *abiS* deletion mutant (Fig. 6).

Lines 161 - 163 & Figure 4b: Expressing *tnaA* & *abiS* in an indole positive strain (DH5 alpha WT) is a messy experiment as the authors cannot rule out interference from the intrinsic tryptophanase. Instead, the authors should express in an indole negative strain (*tnaA* knockout) and show that indole can be produced again by complementing with either *tnaA* or *abiS*. This would be the only way to confirm that *abiS* is working in *E. coli*.

Response: Thanks a lot for your good suggestions. We have revised these data in Fig. 5 as suggested.

Line 165 - 166 & Figure 4b: Assuming that the authors are using LB for this experiment, the supernatant concentration of indole in an *E. coli* culture would be between 300 - 500 μ M (LB has around 300 - 500 μ M tryptophan and the conversion rate of tryptophan to indole via TnaA is 1:1). Thus, using 100 μ M tryptophan for supplementation is a strange choice. The authors should comment on their choice!

Response: (Line 178-195) Thanks a lot for your good suggestions. We have revised these experiments and results as suggested (Fig. 5).

Line 166 -168: 100 μM tryptophan is unlikely to be physiologically relevant therefore it is not surprising that the authors found no effect on *A. baumannii*. This does not rule out that tryptophan might be a precursor for indole production via *abiS*. The authors should supplement with tryptophan in the range of 300 - 500 μM .

Response: Thanks a lot for your good suggestions. As previous study showed that indole production by the TnaA in *E. coli* was determined by exogenous tryptophan (Li et al., 2013, Microbiology (Reading) 159:402-410). As LB culture medium contains 0.5-0.6 mM tryptophan, we then cultured the strains in 0.5 \times LB medium and found that exogenous addition of 300 μM tryptophan could not affect the production of indole in the $\Delta\text{tnaA}(\text{abiS})$ strain, but will significantly induce the production of indole in $\Delta\text{tnaA}(\text{tnaA})$, suggesting that tryptophan is not a precursor for the biosynthesis of indole by AbiS.

Figure 4a: The indole production pathway via TnaA generates pyruvate (not acetone).

Response: Thanks a lot for your good suggestions, we have revised this mistake as suggested.

Figure 5: Why was indole added at 5, 10 & 20 μM ? In *E. coli*, up to 500 μM indole is found in the supernatant of cultures growing in LB? Is it 100 times lower in *A. baumannii*?

Response: Thanks a lot for your good suggestions. Previous study showed that indole production by the TnaA in *E. coli* was determined by exogenous tryptophan (Li et al., 2013, Microbiology (Reading) 159:402-410). *E. coli* can produce up to 500 μM , while the indole yield of *A. baumannii* is much lower than that of *E. coli* and is about 10 μM in LB medium. So, we chose 5, 10 & 20 μM indole for this study.

May 29, 2022

Prof. Yinyue Deng
Sun Yat-sen University
Guangzhou 510642
China

Re: Spectrum01027-22R1 (The cell-cell communication signal indole controls the physiology and interspecies communication of *Acinetobacter baumannii*)

Dear Prof. Yinyue Deng:

Link Not Available

Sincerely,

Beile Gao

Journals Department
Reviewer comments:

Reviewer #1 (Comments for the Author):

The authors have addressed all of my suggestions well.

1. L 31: should be "abiS+" since this is not a mutation. Fix throughout.
2. l 31: should be "an Escherichia coli"
3. L 271: add refs 17 and 33.
4. l 24: should be "behavior"
5. l 42: should be "signal, indole, to modulate"
6. l 69: capitalize "Gram" as it is a person's name.

7. l 132 and throughout: PAO1 is not defined as *P. aeruginosa*.

8. l 287: should be: "Indole production by TnaA in *E. coli* is determined by exogenous tryptophan concentrations (40)."

Reviewer #2 (Comments for the Author):

The manuscript has been improved.

Reviewer #3 (Comments for the Author):

My comment: Provided that tryptophan is not the precursor for indole synthesis via AbiS, the authors did not provide details on the alternative precursor?

Author's answer: Thanks a lot for your good suggestions. We have constructed *E. coli* tnaA deletion mutant and grown these strains in 0.5 x LB and a minimal media lacking tryptophan as suggested. As shown in Fig. 5, exogenous addition of tryptophan will not affect the production of indole performed by AbiS, but will significantly induce the indole production performed by TnaA. In addition, Δ tnaA (abiS) produces almost the same amount of indole in the 0.5 x LB and tryptophan deficient minimal media, suggesting that tryptophan will not affect the biosynthesis of indole via AbiS.

My reply: The authors provided evidence that tryptophan is not the precursor (which was not my question) and failed to provide details on the alternative precursor? What is the alternative precursor? This should be straightforward: purify AbiS protein and perform enzymatic assays with a range of potential precursors (amino acids).

My comment: Why was the concentration of 100 μ M chosen for the motility and cytotoxicity assays? Were other concentrations tested? (same questions for line 120). What is the supernatant concentration of indole in *A. baumannii* LB cultures? The authors should ideally measure the indole supernatant concentration and use that concentration for their assays. The authors need to comment and clarify the physiological relevance of their chosen concentration.

Author's answer: We have tested the effects of indole and its derivatives at different concentrations (25 μ M, 50 μ M and 100 μ M), and found that exogenous addition of 0.1 mM indole showed a strong activity. In addition, we have also modified the methods to measure indole concentration in the supernatant of *A. baumannii* and revised the results in Fig. 4b, which showed that the supernatant concentration of indole culture in LB medium is about 10 μ M. We also found that addition of 10 μ M indole is efficient to restore the defective phenotypes in the abiS deletion mutant (Fig. 6).

My reply: indole physiological concentrations for *E. coli* grown in LB are around 0.5 - 1 mM. The authors detected 10 μ M indole only in the supernatant of *A. baumannii*, which is 100 times less than *E. coli*. Is 10 μ M biologically relevant? why? More importantly, the authors detect 10 μ M in the supernatant using the Kovacs reagent. There is a major technical issue here as the detection window of indole using the Kovács Reagent is in the range of 50 - 500 μ M. Thus, Kovacs is usually considered unreliable at indole concentration below 50 μ M and alternative techniques have been described, including MS-based (e.g. GC-MS, MALDI-TOF MS & CDI-MS) and biochemical-based (e.g. hydroxylamine-based indole assay (HIA) & Quantichrom Indole detection assay). If Kovacs is to be used for low μ M concentration, modifications have been described to increase its sensitivity such as changing the supernatant/Kovács ratio (Liu et al, PlosOne, 2017: e0188853) or using concentration columns (Zarkan et al, SciRep, 2020: 10, 11742). The authors opted to use the standard Kovacs protocol (without modifications) and reported measurements of indole as low as 10 μ M, which raises questions about the accuracy of their measurements. These measurements need to be confirmed using an alternative technique or at least a modified Kovacs protocol that is more suitable for low μ M concentration.

My comment: Figure 4a: The indole production pathway via TnaA generates pyruvate (not acetone).

Author's reply: Thanks a lot for your good suggestions, we have revised this mistake as suggested.

My reply: Figure 4a now says Pyruvic acid which is still wrong. Again, the indole production pathway via TnaA generates pyruvate (not acetone and not pyruvic acid)

Staff Comments:

Preparing Revision Guidelines

- Point-by-point responses to the issues raised by the reviewers in a file named "Response to Reviewers," NOT IN YOUR COVER LETTER.
- Upload a compare copy of the manuscript (without figures) as a "Marked-Up Manuscript" file.

- Each figure must be uploaded as a separate file, and any multipanel figures must be assembled into one file.
- Manuscript: A .DOC version of the revised manuscript
- Figures: Editable, high-resolution, individual figure files are required at revision, TIFF or EPS files are preferred

Please return the manuscript within 60 days; if you cannot complete the modification within this time period, please contact me. If you do not wish to modify the manuscript and prefer to submit it to another journal, please notify me of your decision immediately so that the manuscript may be formally withdrawn from consideration by Microbiology Spectrum.

Point-to-point response to reviewers' suggestions

Reviewer comments:

Reviewer #1 (Comments for the Author):

The authors have addressed all of my suggestions well.

1. L 31: should be "abiS+" since this is not a mutation. Fix throughout.

Response: Thanks for your good suggestions. We have changed "abiS" to "AbiS" as suggested.

2. l 31: should be "an *Escherichia coli*"

Response: We have modified it as suggested.

3. L 271: add refs 17 and 33.

Response: We have added the references as suggested.

4. l 24: should be "behavior"

Response: We have modified it as suggested.

5. l 42: should be "signal, indole, to modulate"

Response: We have modified it as suggested.

6. l 69: capitalize "Gram" as it is a person's name.

Response: Thanks for your good comments. We have modified it as suggested.

7. l 132 and throughout: PAO1 is not defined as *P. aeruginosa*.

Response: Thanks for your good comments. We have changed "PAO1" to "*P. aeruginosa* PAO1" as suggested.

8. l 287: should be: "Indole production by TnaA in *E. coli* is determined by exogenous tryptophan concentrations (40)."

Response: Thanks for your good comments. We have modified it as suggested.

Reviewer #2 (Comments for the Author):

The manuscript has been improved.

Response: Thanks for your nice comments.

Reviewer #3 (Comments for the Author):

My comment: Provided that tryptophan is not the precursor for indole synthesis via AbiS, the authors did not provide details on the alternative precursor?

Author's answer: Thanks a lot for your good suggestions. We have constructed *E. coli tnaA* deletion mutant and grown these strains in 0.5 x LB and a minimal media lacking tryptophan as suggested. As shown in Fig. 5, exogenous addition of tryptophan will not affect the production of indole performed by AbiS, but will significantly induce the indole production performed by TnaA. In addition, $\Delta tnaA$ (*abiS*) produces almost the same amount of indole in the 0.5 x LB and tryptophan deficient minimal media, suggesting that tryptophan will not affect the biosynthesis of indole via AbiS.

My reply: The authors provided evidence that tryptophan is not the precursor (which was not my question) and failed to provide details on the alternative precursor? What is the alternative precursor? This should be straightforward: purify AbiS protein and perform enzymatic assays with a range of potential precursors (amino acids).

Response: Thanks a lot for your good suggestions. To find out the potential precursor of indole biosynthesized by AbiS, we purified AbiS and added each amino acid to carry out *in vitro* enzyme activity experiments, the results showed that AbiS could not

directly utilize these amino acids to produce indole under this condition. Therefore, we added each amino acid to MP medium at 200 μ M, and then detected the production of indole in *E. coli* Δ *tnaA*(*abiS*). We found that exogenous addition of Arg, Asp, Cys, Glu, His, Leu, Lys and Val could increase the yield of indole, but Trp could not, suggesting that Trp is neither a direct precursor nor an indirect precursor for indole biosynthesis performed by *AbiS*.

Effect of exogenous addition of each amino acid on indole production of *E. coli* Δ *tnaA*(*abiS*).

My comment: Why was the concentration of 100 μ M chosen for the motility and cytotoxicity assays? Were other concentrations tested? (same questions for line 120).

What is the supernatant concentration of indole in *A. baumannii* LB cultures? The authors should ideally measure the indole supernatant concentration and use that concentration for their assays. The authors need to comment and clarify the physiological relevance of their chosen concentration.

Author's answer: We have tested the effects of indole and its derivatives at different concentrations (25 μ M, 50 μ M and 100 μ M), and found that exogenous addition of 0.1 mM indole showed a strong activity. In addition, we have also modified the methods to

measure indole concentration in the supernatant of *A. baumannii* and revised the results in Fig. 4b, which showed that the supernatant concentration of indole culture in LB medium is about 10 μ M. We also found that addition of 10 μ M indole is efficient to restore the defective phenotypes in the *abiS* deletion mutant (Fig. 6).

My reply: indole physiological concentrations for *E. coli* grown in LB are around 0.5 - 1 mM. The authors detected 10 μ M indole only in the supernatant of *A. baumannii*, which is 100 times less than *E. coli*. Is 10 μ M biologically relevant? why? More importantly, the authors detect 10 μ M in the supernatant using the Kovacs reagent. There is a major technical issue here as the detection window of indole using the Kovács Reagent is in the range of 50 - 500 μ M. Thus, Kovacs is usually considered unreliable at indole concentration below 50 μ M and alternative techniques have been described, including MS-based (e.g. GC-MS, MALDI-TOF MS & CDI-MS) and biochemical-based (e.g. hydroxylamine-based indole assay (HIA) & Quantichrom Indole detection assay). If Kovacs is to be used for low μ M concentration, modifications have been described to increase its sensitivity such as changing the supernatant/Kovács ratio (Liu et al, PlosOne, 2017: e0188853) or using concentration columns (Zarkan et al, SciRep, 2020: 10, 11742). The authors opted to use the standard Kovacs protocol (without modifications) and reported measurements of indole as low as 10 μ M, which raises questions about the accuracy of their measurements. These measurements need to be confirmed using an alternative technique or at least a modified Kovacs protocol that is more suitable for low μ M concentration.

Response: Thanks a lot for your good suggestions. According to your suggestion, we chose LC-MS/MS to detect the indole production. The detection method was mainly

based on the previous description with a slight modification (Lai Y, Liu CW, Chi L, Ru H, Lu K. 2021. High-Resolution Metabolomics of 50 Neurotransmitters and Tryptophan Metabolites in Feces, Serum, and Brain Tissues Using UHPLC-ESI-Q Exactive Mass Spectrometry. ACS omega 6: 8094–8103.). The detection range of the standard curve is 1-500 μ M.

My comment: Figure 4a: The indole production pathway via TnaA generates pyruvate (not acetone).

Author's reply: Thanks a lot for your good suggestions, we have revised this mistake as suggested.

My reply: Figure 4a now says Pyruvic acid which is still wrong. Again, the indole production pathway via TnaA generates pyruvate (not acetone and not pyruvic acid)

Response: Thanks a lot for your good suggestions. We have deleted Fig 5a as this result is cited from the previous study.

June 16, 2022

Prof. Yinyue Deng
Sun Yat-sen University
Guangzhou 510642
China

Re: Spectrum01027-22R2 (The cell-cell communication signal indole controls the physiology and interspecies communication of *Acinetobacter baumannii*)

Dear Prof. Yinyue Deng:

Your manuscript has been accepted, and I am forwarding it to the ASM Journals Department for publication. You will be notified when your proofs are ready to be viewed.

Sincerely,

Beile Gao
Editor, Microbiology Spectrum
